# Construction of Porcine Epidemic Diarrhea Virus-Like Particles and Its Immunogenicity in Mice

**DOI:** 10.3390/vaccines9040370

**Published:** 2021-04-11

**Authors:** Jihee Kim, Jaewon Yoon, Jung-Eun Park

**Affiliations:** 1Laboratory of Veterinary Public Health, College of Veterinary Medicine, Chungnam National University, Daejeon 34134, Korea; wlgml0721@naver.com (J.K.); hopewj@naver.com (J.Y.); 2Research Institute of Veterinary Science, Chungnam National University, Daejeon 34134, Korea

**Keywords:** porcine epidemic diarrhea, virus-like particle, vaccine, immunogenicity

## Abstract

Porcine epidemic diarrhea (PED), a highly contagious and lethal enteric disease in piglets, is characterized by diarrhea, vomiting, and dehydration, with high mortality in neonatal piglets. Despite the nationwide use of attenuated and inactivated vaccines, the outbreak of PED is still a major problem in the swine industry. Virus-like particles (VLPs) are artificial nanoparticles similar to viruses that are devoid of genetic material and are unable to replicate. VLPs have good safety profiles and elicit robust cellular and humoral immune responses. Here, we generated PED VLPs in eukaryotic cells and examined their immune responses in mice. We found that the M protein is essential for the formation of PED VLPs. Interestingly, PED VLP formation was decreased in the presence of E proteins and increased in the presence of N proteins. Both IgG and IgA antibodies were induced in mice immunized with PED VLPs. Moreover, these antibodies protected against PED virus infection in Vero cells. PED VLPs immunization induced Th2-dominant immune responses in mice. Our results indicate that PED VLPs induce strong immune responses in mice, suggesting that the VLP-based vaccine is a promising vaccine candidate.

## 1. Introduction

Porcine epidemic diarrhea (PED) is a swine enteric disease caused by the PED virus (PEDV) [1]. PED is characterized by watery diarrhea, vomiting, anorexia, dehydration, and weight loss [2]. It affects pigs of all ages, but most severely neonatal piglets, reaching morbidity and mortality of up to 100% [3]. Classical strains of PEDV (G1 strains) were first detected in the UK in 1971, which gradually spread to Asia and Europe [4,5,6,7]. More recently, highly virulent strains (G2 strains) that emerged in China have spread to other countries, including the United States [8,9].

PEDV is a member of the genus Alphacoronavirus in the Coronaviridae family [1]. The viral genome is single-stranded RNA (28,000 nucleotides in length) of positive-sense polarity containing 3′- and 5′-untranslated regions (UTR) at both ends [1,10]. Two-thirds of the genome from the 5′-end encode proteins necessary for RNA replication [11]. The one-third on the 3′-end of the genome comprises at least seven open reading frames (ORFs) that include four structural proteins, namely S (spike), E (envelope), M (membrane), and N (nucleocapsid) and accessory protein, ORF3 [1,10,11]. The S protein mediates attachment of the virus to the host cell surface receptors and subsequent fusion between the viral and host cell membranes to facilitate viral entry into the host cell [12,13]. It is a key target for the host antibody response and a good candidate for a protein-based vaccine immunogen [14]. The M protein is the most abundant structural protein, which adapts a part of the membrane for viral assembly and captures other structural proteins at the budding site [15,16]. The E protein is found in small amounts within the virion and helps in viral assembly and release [17,18,19]. The N protein packages the viral genome, and the association of N protein with the ER-Golgi complex plays a role in viral budding [20,21].

One of the most important methods for controlling PED outbreaks is vaccination. Both attenuated and inactivated PEDV vaccines have been routinely used in Asian countries for several years. Vaccination of sows prior to farrowing induces lactogenic immunity, which is transferred to neonatal piglets via colostrum [22]. Inactivated vaccines are safe, but have a short duration of immunity and require the appropriate adjuvants for strong immune responses [23]. Live-attenuated vaccines, produced by serially passaging field strains, are more effective against homologous strains but have a long lead development time and have been associated with safety concerns of recombination with field strains [24]. Regardless of the type of vaccine used, viremia and transmission of PEDV are not fully prevented in vaccinated animals [25,26,27,28]. Outbreaks in vaccinated herds and the periodical emergence of new, highly pathogenic variants continue in countries where vaccines have been routinely used for many years [27,29].

Virus-like particles (VLPs) are composed of viral structural proteins, and their morphologies are similar to those of the original virus. Since VLPs lack genetic material, there are no risks of reversion to virulence [30,31]. VLPs can induce robust immune responses compared with inactivated or live-attenuated virus vaccines [30,31]. In this study, we generated PED VLPs and examined their immunogenicity in mice. Our data suggest that VLPs are immunologically effective as a PED vaccine.

## 2. Materials and Methods

### 2.1. Cells and Viruses

African green monkey kidney cells (Vero, CCL-81) were grown in Dulbecco’s Modified Eagle’s Medium (DMEM, HyClone Laboratories Inc., South Logan, UT, USA) supplemented with 10% fetal bovine serum (FBS, HyClone Laboratories Inc., South Logan, UT, USA), 100 IU/mL penicillin (Gibco, Waltham, MA, USA), and 100 μg/mL streptomycin (Gibco, Waltham, MA, USA). HEK293T cells were grown in DMEM supplemented with 10% FBS, 100 IU/mL penicillin (Gibco, Waltham, MA, USA), 100 μg/mL streptomycin (Gibco, Waltham, MA, USA), and 100 μg/mL HEPES (Gibco, Waltham, MA, USA). Cells were maintained at 37 °C and CO_2_. Cell culture materials and reagents were obtained from SPL Life Sciences Co., Ltd. (Gyeonggi-do, Korea) and Hyclone (HyClone Laboratories Inc., South Logan, UT, USA) unless otherwise stated.

Vero cell-adapted PEDV strain SM98 was propagated in Vero cells, as described previously [32].

### 2.2. Plasmids

Human codon-optimized sequences of genes encoding S, M, E, and N structural proteins of PEDV (GenBank: AF353511) with C-terminal C9 tag (for S) or Myc tag (for M, E, and N) were synthesized by Genscript Biotechnology. *EcoRI* and *XhoI* restriction sites were placed at the 5′- and 3′-ends, respectively. The four genes were cloned into the double *EcoRI* and *XhoI* restriction sites of the expression vector pCAGGS. The correct orientation of the insertions was examined using restriction digestion analysis and DNA Sanger sequencing.

### 2.3. Production and Purification of VLPs

HEK293T cells were transfected with plasmids encoding PEDV structural proteins as indicated. For transfection of a single plasmid, 2 µg of each plasmid was applied individually. Co-transfection of multiple plasmids was conducted with an equal molar of each plasmid in a total of 2 µg. Transfection was performed by incubating plasmid DNAs with polyethylenimine (PEI; Polysciences) at 1:3 DNA:PEI ratios in Opti-MEM (Life Technologies, Carlsbad, CA, USA) for 15 min at 25 °C. Cell-free supernatants containing the nanoparticles were collected at 48 h post-transfection and filtered through 0.45-µm syringe filters (Port Washington, NY, USA). VLPs were concentrated by centrifugation at 100,000× *g* for 3 h with 20% sucrose cushion, suspended in phosphate-buffered saline (PBS) to a 100-fold concentration, and stored at −80 °C until use.

### 2.4. Western Blot Analysis

Concentrated VLPs and infected cell lysates were mixed with SDS solubilizer to final concentrations of 0.0625 M Tris-HCl (pH 6.8), 10% glycerol, 0.01% bromophenol blue, 2% (*w*/*v*) SDS, and 1% 2-mercaptoethanol. Samples were heated at 95 °C for 5 min, separated in 10% (*w*/*v*) polyacrylamide-SDS gels, transferred to PVDF membranes, probed with monoclonal mouse anti-C9 (Santa Cruz) or monoclonal mouse anti-Myc antibody (Santa Cruz, Dallas, TX, USA). Membranes were then probed with horseradish peroxidase-conjugated goat anti-mouse IgG (Bioss, Woburn, MA, USA), developed with ECL substrate (Thermo Fisher Scientific, Middlesex, MA, USA), and signals were detected using Fusion Solo X (Vilber, France). Band density on western blot membranes was analyzed using Evolution Capt software (Vilber, France) (shown in Appendix A).

### 2.5. Mice Immunization

The animal experiments were performed according to the protocol approved by the Institutional Animal Care and Use Committee of Chungnam National University (ethics approval number: CNU-01184).

Eight-week-old female BALB/C mice were purchased from Samtaco (Gyeonggi-do, Korea) and were divided into three groups of five mice each. Mice were intraperitoneally immunized with 50 μg of PED VLPs, UV-inactivated PEDV, or PBS with alum adjuvant. For UV-inactivated PEDV, SM98 was inactivated using UV irradiation for 30 min 25 °C, concentrated by ultracentrifugation (100,000× *g* for 3 h at 4 °C), and suspended in PBS. Vaccinations were performed twice with a 2-week interval. At 28 days post-initial immunizations, serum samples were collected by cardiac puncture.

### 2.6. Enzyme-Linked Immunosorbent Assay (ELISA)

Antibody titers of PEDV-specific IgG and IgA in sera from immunized mice were determined using ELISA, as described previously by Park et al. [33]. Briefly, microtiter plates were coated with 100 μL of SM98 (10^5^ TCID50/mL) overnight at 4 °C and blocked with 5% skim milk for 1 h at 25 °C. Diluted samples were added and kept for 1 h, followed by incubation for 1 h with HRP-conjugated goat anti-mouse IgG, IgG1, IgG2a, or IgA antibodies (Bethyl Laboratories, Montgomery, TX, USA). The enzymatic activity was detected by adding 3, 3′, 5, 5′-tetramethylbenzidine substrate. Then, the reaction was stopped with 2N H_2_SO_4_, and the absorbance at 450 nm was measured on a microplate reader (PerkinElmer, Waltham, MA, USA).

### 2.7. Serum Neutralization (SN) Test

The Serum Neutralization (SN) test was performed as described previously by Park et al. [33]. SN titers were expressed as the reciprocals of the highest serum dilution resulting in the inhibition of the cytopathic effect.

### 2.8. Statistical Analyses

All experiments except animal experiments were independently repeated at least three times. Data are presented as the mean ± SD. Statistical analysis was performed using the Holm–Sidak multiple Student’s *t*-test. A *p*-value of < 0.05 was considered statistically significant.

## 3. Results

### 3.1. M Protein Expression Is Sufficient for the Formation of PED VLPs

The PEDV S, E, M, and N genes were cloned into the pCAGGS vector, and the recombinant plasmid was confirmed using restriction digestion analysis as well as DNA sequencing. We first examined the secretory features of the four PEDV structural proteins. HEK293T cells were transfected with plasmids expressing S, E, M, or N genes. Western blotting analysis using cell lysates revealed the expression of four structural proteins; the S, E, M, and N proteins were detected as single bands of 200, 15, 24, and 55 kDa, respectively (Figure 1A and Appendix A). As shown in Figure 1A, the M protein could be easily released into the medium (supernatant) independent of other structural proteins. The S, E, and N proteins were least detectable in the culture supernatant.

To better understand the role of four PEDV structural proteins in viral egress and VLP formation, we co-transfected HEK293T cells with plasmids expressing S, E, M, or N in various combinations (Figure 1B and Appendix A). In cells transfected with S only or S and N proteins, no S proteins were detected in the supernatant. Whenever M proteins were present, S proteins were detected in the supernatant. Additionally, the expression of E proteins decreased the amount of S proteins in the supernatants. Conversely, the additional expression of N proteins increased the amount of S proteins in the supernatants. These results demonstrate that PED VLPs autonomously assemble in mammalian cells and that the M protein is essential for PED VLP formation.

### 3.2. PED VLPs Induce Strong Immune Responses in Mice

To determine the immunogenicity of PED VLPs, BALB/C mice were immunized with 50 μg of PED VLPs containing S and M proteins, PED VLPs containing S, M, and N proteins, UV-inactivated PEDV, or PBS (Figure 2A). We assessed serum IgG and IgA titers against PEDV using ELISA. IgG titers were significantly higher in mice vaccinated with inactivated PEDV compared to VLP vaccine groups (Figure 2B). IgA titers were significantly higher in mice vaccinated with VLPs containing S, M, and N proteins compared to inactivated PEDV vaccine group (Figure 2C). Data indicate that PED VLP immunization induced humoral immune responses in mice.

To determine whether Th1 and/or Th2 immune responses were induced by vaccination with PED VLPs, the levels of specific IgG1 and IgG2a subclasses were measured. Both IgG1 titers were higher in PED VLP-immunized mice than in UV-inactivated PEDV-immunized mice (Figure 3A). The IgG2a titers of the three groups were similar (Figure 3B). Analysis of the antigen-specific IgG1/IgG2a ratio revealed that PED VLP immunization induced a Th2-dominant immune response in mice (Figure 3C).

### 3.3. PED VLP Immunization Induced Neutralizing Antibodies

Neutralizing antibodies play a major role in protecting against PEDV infection [34]. Therefore, we examined the SN activity against PEDV (Figure 4). SN titers of sera from PED VLP-immunized mice were higher than those from UV-inactivated PEDV-immunized mice. There were no differences in neutralizing activity between PED VLPs containing S and M proteins and those containing S, M, and N proteins. The data indicate that PED VLPs efficiently induce SN antibodies in mice.

## 4. Discussion

The outbreak of PEDV has become a global concern in the swine industry. Current vaccines do not fully protect against PEDV infection. As a new vaccine platform for PED, we generated PED VLPs and examined their immunogenicity in mice. Our results demonstrated that PED VLPs autonomously assembled in mammalian cells and that the M protein was essential for PED VLP formation. VLP immunization in mice induced strong humoral immune responses in mice and conferred neutralizing activity against PEDV.

The components of VLPs have been studied in many coronaviruses. The requirements of the structural proteins differ depending on the viruses and expression systems. Previous studies of mouse hepatitis virus (MHV) reported that E and M proteins are essential for the assembly of virus particles and co-expression of E and M proteins in cultured cells produced VLPs that are not infectious [35]. In SARS-CoV, M and E proteins are required for VLP formation and N proteins are needed for the assembly of viral RNA [36,37]. MERS-CoV VLPs were also produced in insect cells expressing M and E proteins [38]. More recently, it was shown that the expression of M and E proteins is essential for the efficient formation and release of SARS-CoV-2 VLPs [39]. Wang et al. produced PED VLPs composed of S, M, and E proteins with a baculovirus expression system [40]. Here, we demonstrated the role of PEDV structural proteins in viral egress and VLP formation in mammalian cells. We found that the M protein is sufficient for the formation of PED VLPs in the mammalian expression system. PED VLP formation was enhanced by the additional expression of N proteins but was decreased by the additional expression of E proteins. Collectively, when the cells co-expressed M and N proteins, VLP formation was the highest.

One of the strengths of VLP-based vaccines is safety. Due to the lack of genetic materials that determine the pathogenicity of viruses, VLP is non-infectious and can be handled in normal laboratory settings without biosafety protection. Therefore, VLP constitutes a safe and relevant model in molecular studies of viral entry and virion egress, as well as a vaccine candidate [41]. Live-attenuated vaccines can cause the acquisition of pathogenicity and can lead to the emergence of variant viruses with continuous use [24]. Although the inactivated vaccine has higher safety than the live-attenuated vaccine, there is a concern that safety may deteriorate due to incomplete inactivation [23]. The development of a VLP-based vaccine could overcome this safety issue.

Besides safety issues, traditional vaccine platforms have limitations in that virus adaptation is required for vaccine manufacture. To increase viral titers and/or decrease their pathogenicity, isolated viruses are continually passaged in cultured cells, such as Vero cells. The process of virus adaptation usually takes a long time. In addition, during passaging, viruses obtain cell-adapted mutations in the gene encoding the S protein, which can affect the antigenicity and protective efficacy of vaccines [33]. As VLP-based vaccines are developed by molecular technology, researchers can respond immediately to the emergence of variant strains and produce vaccines with the same antigenicity profile as the wild-type viruses.

Here, we compared the immune responses of VLPs containing S and M proteins and VLPs containing S, M, and N proteins. IgA titer and SN activity were high in mice immunized with VLPs containing S, M, and N proteins (Figure 2 and Figure 4). We speculate that the morphology of VLP might be more rigid and similar to authentic viruses when the N protein is present. Thereby, the additional expression of the N protein can facilitate the immunogenicity of VLPs as well as VLP formation [21,36,42]. Furthermore, it is suggested that the N protein can elicit a broad-spectrum cellular immune response [21,43]. Taken together, our data show that the N protein promotes immunogenicity as well as VLP formation.

The use of adjuvants increases the side effects of vaccines. Therefore, we tested whether adjuvants can be excluded from VLP vaccination (data not shown). Unfortunately, VLP immunization did not induce strong immune responses in the absence of adjuvants. It is necessary to overcome these limitations by developing a VLP vaccine that co-expresses a protein that can enhance the immunogenicity of the vaccine.

In order to evaluate the vaccine efficacy, it is necessary to evaluate its defense against virus challenge. However, in this study, the immunogenicity in mice was evaluated, and the mice were not susceptible to PEDV, so the protective efficacy against virus challenge was not evaluated. Although the current study cannot confirm the protection against virus challenge, it provides the preclinical evaluation that the VLP vaccine can effectively induce an immune response. In addition, since neutralizing antibodies play an important role in PEDV defense, the fact that the VLP vaccine induces higher neutralizing antibodies than inactivated PEDV vaccines indirectly proves the efficacy of the vaccine.

## 5. Conclusions

The primary advantages of this innovative approach are safety, efficacy, ease of handling, and a short development time. As the method can be easily adapted to newly evolving strains, provided they are readily cultured, this approach is very relevant to current field immunization practices of feedback exposure and autogenous vaccination. Our future goals include testing the VLP vaccine in pregnant sows and piglets and improving oral and respiratory mucosal vaccine delivery systems to confer protection.

## Figures and Tables

**Figure 1 vaccines-09-00370-f001:**
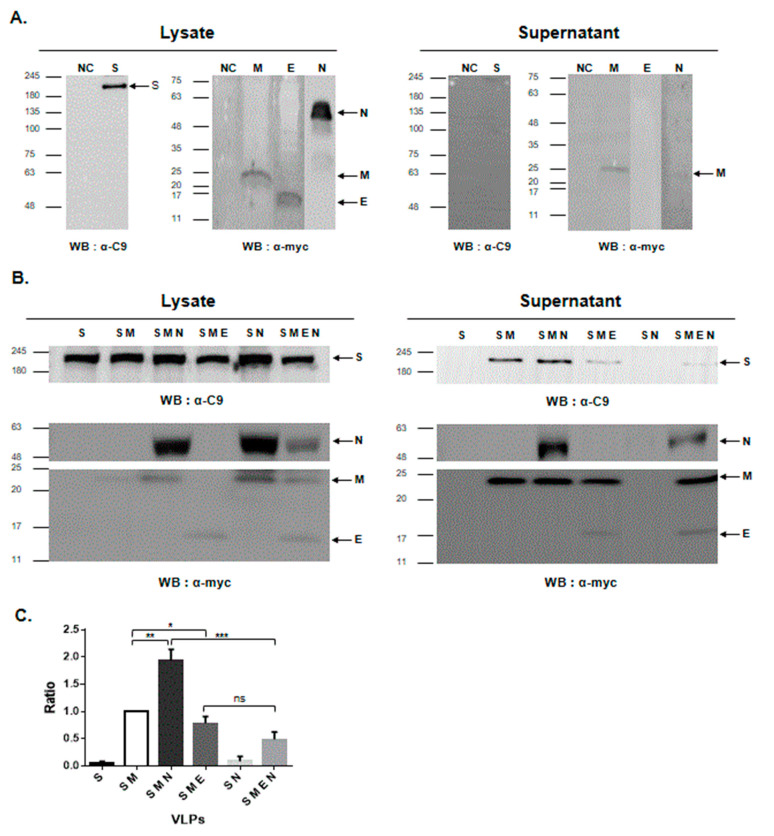
Generation of porcine epidemic diarrhea virus-like particles (PED VLPs) in mammalian expression systems. (**A**,**B**) HEK293T cells were transfected with the indicated plasmids singly (**A**) or in combinations (**B**). At 48 h post-transfection, the expression of four structural proteins in the cell lysates and supernatants were analyzed using western blotting. The numbers at the left indicate molecular mass in kilodaltons. S, E, M, and N proteins are indicated. (**C**) Band intensities of the S protein in western blot from supernatants in (**B**) were measured and plotted relative to those in cells transfected with S and M proteins. Statistical significance was assessed using Student’s *t*-test. *, *p* < 0.05; **, *p* < 0.01; ***, *p* < 0.001; ns, not significant.

**Figure 2 vaccines-09-00370-f002:**
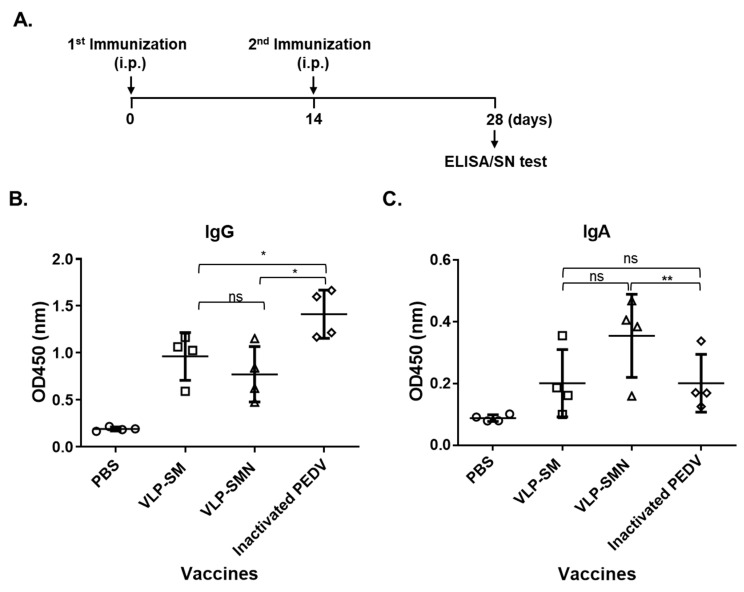
Humoral immune response induced by PED VLPs. (**A**) Schematic diagram of the immunization protocol. BALB/c mice were i.p. immunized with the indicated PED VLPs (square, triangle), UV-inactivated PEDV (diamond), or PBS (circle) at 0 and 14 days after the first immunization. At 28 days post-immunization, serum samples were collected. (**B**,**C**) The levels of PEDV-specific IgG (**B**) and IgA (**C**) were measured by ELISA. Results are expressed as the mean (*n* = 3) at OD450 ± SD values and are representative of at least two independent experiments. Statistical significance was assessed using Student’s *t*-test. *, *p* < 0.05; **, *p* < 0.01; ns, not significant.

**Figure 3 vaccines-09-00370-f003:**
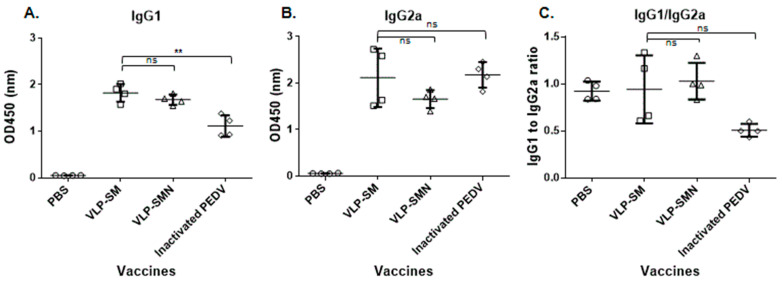
Influence of PED VLP immunization on the type of T helper cell response. BALB/c mice were i.p. immunized with the indicated PED VLPs (square, triangle), UV-inactivated PEDV (diamond), or PBS (circle), and serum samples were collected at 28 days post-immunization. The serum-specific IgG1 (**A**) and IgG2a (**B**) titers were determined as indicators of a Th2 or Th1 type of response, respectively. (**C**) IgG1/IgG2a titer ratios of immunized mice. Results are expressed as the mean (*n* = 4) at OD450 ± SD values and are representative of at least two independent experiments. Statistical significance was assessed using Student’s *t*-test. **, *p* < 0.01; ns, not significant.

**Figure 4 vaccines-09-00370-f004:**
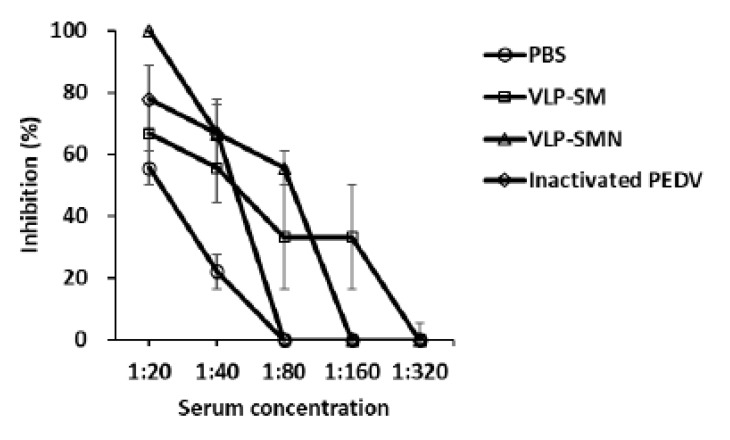
Protective efficacy of PED VLP immunization. Detection of neutralizing antibody titers in the serum of the immunized mice using the serum neutralization (SN) test. Results are expressed as the mean (*n* = 4) of SN titers ± SD values and are representative of at least two independent experiments.

## Data Availability

Data sharing is not applicable to this article.

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
