# Peer review of "Construction of Porcine Epidemic Diarrhea Virus-Like Particles and Its Immunogenicity in Mice"

_vaccines, 2021, doi:10.3390/vaccines9040370_

Round 1

Reviewer 1 Report

The manuscript by Kim et al. generated a PEDV Virus-like particles (VLPs) and found viral M protein is essential for VLPs formation. Moreover, in presence of viral E proteins decreased VLPs formation, but with viral N protein increased VLPs formation. Further immunized mice with VLPs could induce the Th2-dominant immune responses in mice and IgG and IgA antibodies, indicating that PEDV VLPs-based vaccine might be a promising candidate.

Although PEDV VLPs generation is very interesting, but the main weakness of this work presented in this manuscript is that authors did not firmly prove the VLPs assembly is correct. This is due to the fact that authors did not perform a morphological evaluation of this VLPs.

Major comments:

  1. As the formation of VLPs is essential for this story, the morphological evaluation of this VLPs must be done to confirm it. Moreover, Kim et al. indicate the production and purification of VLPs in the Materials and Methods, but the 100,000xg centrifugation only means an enrichment and purification should further move to sucrose-gradient ultracentrifugation.

  1. For the mice immunization, Kim et al. used 50 µg VLPs, it might be good to include a range of quantities. This will be good to know which quantity of VLPs immunization is effective.

  1. As we know, coronaviral N protein could form an oligomer and M protein interacts with N protein, E protein and S protein, there should be a ratio between those structural proteins for VLPs formation. That could also be the reason why in presence of viral E protein could decrease VLPs formation and N protein increase the formation. So the author should first verify the ratio between structural proteins, otherwise this will affect the formation of VLPs.

Author Response

Reviewer 1

The manuscript by Kim et al. generated a PEDV Virus-like particles (VLPs) and found viral M protein is essential for VLPs formation. Moreover, in presence of viral E proteins decreased VLPs formation, but with viral N protein increased VLPs formation. Further immunized mice with VLPs could induce the Th2-dominant immune responses in mice and IgG and IgA antibodies, indicating that PEDV VLPs-based vaccine might be a promising candidate.

Although PEDV VLPs generation is very interesting, but the main weakness of this work presented in this manuscript is that authors did not firmly prove the VLPs assembly is correct. This is due to the fact that authors did not perform a morphological evaluation of this VLPs.

Major comments:

  1. As the formation of VLPs is essential for this story, the morphological evaluation of this VLPs must be done to confirm it. Moreover, Kim et al. indicate the production and purification of VLPs in the Materials and Methods, but the 100,000xg centrifugation only means an enrichment and purification should further move to sucrose-gradient ultracentrifugation.

Response:To analyze morphological evaluation of VLPs, electron microscopy (EM) had to be conducted. Unfortunately, we are in a situation where we are unable to perform EM analysis. Since the PEDV S protein is not naturally secreted from the transfected cells, we believe that PED VLPs were formed when the S protein could be detected in the supernatants.

We performed ultracentrifugation of VLPs with 20% sucrose in the VLPs purification process. We revised our manuscript in line 97.

  1. For the mice immunization, Kim et al. used 50 µg VLPs, it might be good to include a range of quantities. This will be good to know which quantity of VLPs immunization is effective.

Response:Thank you for your comments. However, the current experiment was performed under single dose conditions. In the future experiments, we will perform experiments with various quantity of VLPs.

  1. As we know, coronaviral N protein could form an oligomer and M protein interacts with N protein, E protein and S protein, there should be a ratio between those structural proteins for VLPs formation. That could also be the reason why in presence of viral E protein could decrease VLPs formation and N protein increase the formation. So the author should first verify the ratio between structural proteins, otherwise this will affect the formation of VLPs.

Response:We found that E and N protein expression regulates PED VLP formation differently. However, we are not entirely sure why and how these proteins regulate VLP formation. It is interesting that the natural composition of structural proteins may be relevant to our observations. However, when we generated VLPs from cells transfected with structural proteins at different compositions, we did not see a significant difference.

Reviewer 2 Report

Kim et al. construcyed PED VLPs in HEK239 cells and examined the immune responses of VLPs in mice. They found that M protein is essential for the formation of PED VLPs and the VLPs can induce IgG and IgA antibodies.  They hypotheized that PED VLPs immunization induced Th2-dominant immune responses in mice. Also, they examined the neutralizing activity of the induced antibodies in vero cells.

The production of VLPs from PEDV in different systems such as insects has been reported before. But the authors provide bew findings about the importance of M protein in the production of PEDV VLPS. However, I have several major concerns about the manuscript

1- The manuscript needs language editing

2- The number of mice in Figure2 is low (3 mice/ group), this low number of mice is not sufficient to have a solid conclusion. The authors need to increase the number of mice at least 5 mice/ group.  

3- Figure legend of Figure 2 mentioned that " Results are expressed as the mean (n = 3) at OD450 ± SD values and are representative of at least two independent experiments."

How can 3 mice/ group come from 2 different independent experiments?

3- The number of mice in Figure 3 is different than the number of mice in Figure 2. Is this different experiment/ different animals??????

In general In vivo results need revision and good design from the authors

4- The authors claimed that VLPs induce Th2 immune response by measurement of IgG1, IgG2, but this is not convinicing. The authors should measure cytokines associated with each Th subtypes such as IL-4, IL-`12, IFN-g before claim this finding.

5- To assess the neutalization activity in Figure 4, the authors should provide IF images/ other confirmation from for the inhibition activity of the induced antibodies.

Author Response

Reviewer 2

Kim et al. construcyed PED VLPs in HEK239 cells and examined the immune responses of VLPs in mice. They found that M protein is essential for the formation of PED VLPs and the VLPs can induce IgG and IgA antibodies.  They hypotheized that PED VLPs immunization induced Th2-dominant immune responses in mice. Also, they examined the neutralizing activity of the induced antibodies in vero cells.

The production of VLPs from PEDV in different systems such as insects has been reported before. But the authors provide bew findings about the importance of M protein in the production of PEDV VLPS. However, I have several major concerns about the manuscript

1- The manuscript needs language editing
Response: The manuscript received language editing service provided by Elsevier. However, if you need additional language editing, we will be happy to do it.

2- The number of mice in Figure2 is low (3 mice/ group), this low number of mice is not sufficient to have a solid conclusion. The authors need to increase the number of mice at least 5 mice/ group.  
Response: For immunization, 5 mice per group were used. Since serum samples were not obtained from a few mice, 4 serum samples were used for analysis (Figure 2, 3, and 4). In Figure 2, although 4 mice per group were used, outliers occurred and data was removed. However, to reduce the confusion of the reader, the Figure 2 was changed to contain data from 4 animals.

3- Figure legend of Figure 2 mentioned that " Results are expressed as the mean (n = 3) at OD450 ± SD values and are representative of at least two independent experiments." How can 3 mice/ group come from 2 different independent experiments?
Response: Two independent experiments were carried out, and the results obtained from one experiment were shown as a figure.

4- The number of mice in Figure 3 is different than the number of mice in Figure 2. Is this different experiment/ different animals??????
Response: For immunization, 5 mice per group were used. Since serum samples were not obtained from a few mice, 4 serum samples were used for analysis (Figure 2, 3, and 4).

In general In vivo results need revision and good design from the authors
Response: Unfortunately, we do not have enough time to perform the in vivo experiment for this revision. However, we carefully went through the manuscript and revised it.

4- The authors claimed that VLPs induce Th2 immune response by measurement of IgG1, IgG2, but this is not convinicing. The authors should measure cytokines associated with each Th subtypes such as IL-4, IL-`12, IFN-g before claim this finding.
Response: The IgG1 isotype is generally associated with Th2 humoral immune responses, whereas higher levels of IgG2a indicate a Th1 cellular immune response. Therefore, the ratio of these two isotypes indicates the type of immune response generated against a given antigen (Ferreira et al. 2008). Although cytokine analysis can provide more information against immune responses, we did not have enough amounts of serum samples to perform it. We could perform the in vivo experiment to measure cytokine expression in serum samples, however we do not have enough time to perform the in vivo experiment for this revision.

5- To assess the neutalization activity in Figure 4, the authors should provide IF images/ other confirmation from for the inhibition activity of the induced antibodies.
Response: For the evaluation of neutralization activity, we used TCID50 analysis, and immunostaining using fluorescent antibodies was not performed.

Reviewer 3 Report

This manuscript attempted a better alternative vaccine platform for the existent live attenuated or inactivated immunization to Porcine Epidemic Diarrhea. The main concerning addressed in this article was the safety of live attenuated vaccine and the immunogenicity as well as long-lasting memory that is inexistent on inactivated commercial vaccines against PED. To address safety concerning the authors produced in vero cells a virus-like particles with different PED proteins and evaluated if these PED VLP induced antibodies and if the antibodies neutralized PED. The authors finding that PED VLP induced Th2 immune response with IgG and IgA antibodies in mice and presented neutralizing activity.

Major: The manuscript presents an error in the IgG1 and IgG2a antibody response. I suggested that the authors have additional data to support evidence about Porcine Epidemic Diarrhea immune response. It would be beneficial for the quality of this manuscript to add data (cytokine secretion) to address whether the immune response is antibody or cell dependent.    

Minor:

  1. Immunization IP mice with PED VLPs, UV-inactivated PEDV, or PBS. In the methods is unclear if the there was a control group that tested PBS only and another control group that tested PBS+ Adjuvant? Please clarify that.
  2. The authors stated that: “IgG titers were robust in all vaccinated mice. IgA titers were significantly higher in mice vaccinated with VLPs containing S, M, and N proteins”. However, in the Fig 2: Considering that mice were immunized with VLP-SM; VLP-SMN or inactivated PEDV then is noticed that IgG title was high only on the inactivated PEDV group and IgA title was high in VLP-SMN.
  3. Fig3: Misconception on legend: IgG1 induces to Th2 and IgG2a induces to Th1 immune response.
  4. Fig 3: Please put all the graphics on the same scale.
  5. The authors stated that “Analysis of the antigen-specific IgG1/IgG2a ratio revealed that PED VLP immunization induced a Th2-dominant immune response in mice”. This statement does not reflect the data. Please re analyze the data.
  6. In the introduction, the authors stated that: “Regardless of the type of vaccine used, viremia and transmission of PEDV are not fully prevented in vaccinated animals” why you believe that your subunit proteic vaccine will protect pigs, there was no challenge on this study, therefore, more date is necessary to accomplish this statement.
  7. There is some distracting information (lines 34-35)
  8. Line 189-190: Both IgG1 and IgG2a titers were higher in PED VLP-immunized mice than in UV-inactivated PEDV-immunized mice (Figure 3A and 3B). However, according to the figure 3B there was no difference of IgG2a titer for the three groups of PEDV-immunized mice.
  9. Line 191-192: IgG1 induces to Th2 and IgG2a induces to Th1 immune response. Then the statement is incorrect “Please read reference: Germann, T., M. Bongartz, H. Dlugonska, H. Hess, E. Schmitt, L. Kolbe, E. Kölsch, F. J. Podlaski, M. K. Gately, and E. Rüde. 1995. Interleukin-12 profoundly up-regulates the synthesis of antigen-specific complement-fixing IgG2a, IgG2b and IgG3 antibody subclasses in vivo J. Immunol.25:823-829.
  10. The authors could discuss more about correlates of protection in pigs PED infected.

Author Response

REReviewer 3

This manuscript attempted a better alternative vaccine platform for the existent live attenuated or inactivated immunization to Porcine Epidemic Diarrhea. The main concerning addressed in this article was the safety of live attenuated vaccine and the immunogenicity as well as long-lasting memory that is inexistent on inactivated commercial vaccines against PED. To address safety concerning the authors produced in vero cells a virus-like particles with different PED proteins and evaluated if these PED VLP induced antibodies and if the antibodies neutralized PED. The authors finding that PED VLP induced Th2 immune response with IgG and IgA antibodies in mice and presented neutralizing activity.

Major: The manuscript presents an error in the IgG1 and IgG2a antibody response. I suggested that the authors have additional data to support evidence about Porcine Epidemic Diarrhea immune response. It would be beneficial for the quality of this manuscript to add data (cytokine secretion) to address whether the immune response is antibody or cell dependent.
Response: The IgG1 isotype is generally associated with Th2 humoral immune responses, whereas higher levels of IgG2a indicate a Th1 cellular immune response. Therefore, the ratio of these two isotypes indicates the type of immune response generated against a given antigen (Ferreira et al. 2008). Although cytokine analysis can provide more information against immune responses, we did not have enough amounts of serum samples to perform it. We could perform the in vivo experiment to measure cytokine expression in serum samples, however we do not have enough time to perform the in vivo experiment for this revision.

Minor:

  1. Immunization IP mice with PED VLPs, UV-inactivated PEDV, or PBS. In the methods is unclear if the there was a control group that tested PBS only and another control group that tested PBS+ Adjuvant? Please clarify that.

Response: The PBS group that we vaccinated was the PBS with adjuvants. There was no significant difference in titer values when inoculated with PBS with and without adjuvant. To clarify that, we revised our manuscript in line 116.

  1. The authors stated that: “IgG titers were robust in all vaccinated mice. IgA titers were significantly higher in mice vaccinated with VLPs containing S, M, and N proteins”. However, in the Fig 2: Considering that mice were immunized with VLP-SM; VLP-SMN or inactivated PEDV then is noticed that IgG title was high only on the inactivated PEDV group and IgA title was high in VLP-SMN.

Response: Because IgG titers were high in all vaccinated groups, we described as “IgG titers were robust in all vaccinated mice”. To describe the results more accurately, we revised our manuscript in lines 177-180.

  1. Fig3: Misconception on legend: IgG1 induces to Th2 and IgG2a induces to Th1 immune response.

Response: Thank you for your comments. We revised our manuscript in line 199.

  1. Fig 3: Please put all the graphics on the same scale

Response: Thank you for your comments. We revised Figure 3.

  1. The authors stated that “Analysis of the antigen-specific IgG1/IgG2a ratio revealed that PED VLP immunization induced a Th2-dominant immune response in mice”. This statement does not reflect the data. Please re analyze the data.

Response: The IgG1 isotype is generally associated with Th2 humoral immune responses, whereas higher levels of IgG2a indicate a Th1 cellular immune response. Therefore, the ratio of these two isotypes indicates the type of immune response generated against a given antigen (Ferreira et al. 2008). As IgG1/IgG2a ratio was greater than 1 in VLP vaccinated mice, we concluded that VLP vaccination induced a Th2-dominant immune response.

  1. In the introduction, the authors stated that: “Regardless of the type of vaccine used, viremia and transmission of PEDV are not fully prevented in vaccinated animals” why you believe that your subunit proteic vaccine will protect pigs, there was no challenge on this study, therefore, more date is necessary to accomplish this statement.
    Response: Challenge experiment require specific facilities. Currently, we have no ABL2 facility to do PEDV challenge experiment, so we haven`t been able to conduct this experiments. I think it need to be evaluated in the future.
  2. There is some distracting information (lines 34-35)

Response: Thank you for your comments, however we think that the information described in lines 34-35 is needed. Thank you.

  1. Line 189-190: Both IgG1 and IgG2a titers were higher in PED VLP-immunized mice than in UV-inactivated PEDV-immunized mice (Figure 3A and 3B). However, according to the figure 3B there was no difference of IgG2a titer for the three groups of PEDV-immunized mice.

Response: To describe the results more accurately, we revised our manuscript in lines 192-194.

  1. Line 191-192: IgG1 induces to Th2 and IgG2a induces to Th1 immune response. Then the statement is incorrect “Please read reference: Germann, T., M. Bongartz, H. Dlugonska, H. Hess, E. Schmitt, L. Kolbe, E. Kölsch, F. J. Podlaski, M. K. Gately, and E. Rüde. 1995. Interleukin-12 profoundly up-regulates the synthesis of antigen-specific complement-fixing IgG2a, IgG2b and IgG3 antibody subclasses in vivo Immunol.25:823-829.

Response: The IgG1 isotype is generally associated with Th2 humoral immune responses, whereas higher levels of IgG2a indicate a Th1 cellular immune response. Therefore, the ratio of these two isotypes indicates the type of immune response generated against a given antigen (Ferreira et al. 2008).

  1. The authors could discuss more about correlates of protection in pigs PED infected.

Response: Thank you for your comments. We revised our manuscript in lines 268-275.

Round 2

Reviewer 1 Report

Dear authors, Thanks for your really fast reply.

After going through your comments, first, it is difficult to accept your comments on using S protein secretion as a marker to see whether the VLPs are formed correctly. Because it is super indirect, within your figure 2B, the supernatant one, transfection with only S and M protein is enough to detect the S protein in the supernatant. With S, M and N proteins, you could detect the same amount of S protein in the supernatant. If you think S and M is enough to form a VLP, then what is the function of N protein in VLPs? To confirm it, the EM is the only option and should be done for a good manuscript.

Moreover, I have proposed different doses of VLP immunization in mice, and you indicated you will do it in the future. However, without different doses of VLP immunization, this could be just due to the immunization from any viral proteins, not from the VLPs.

I hope the authors from this manuscript could re-consider to perform some experiments to prove the weakness of this story.

Author Response

Dear reviewer,

Thank you for your comments.

Assessing VLP production by detecting S protein in the supernatant is indirect, but it has been used in many studies as a general method indicating VLP production. We agree that EM can help confirm VLP production and improve the quality of the paper. However, I am not sure that EM is suitable for evaluating the amount and structural stability of VLPs. We showed the amounts of VLP through western blot and quantification in Figure 1B and 1C.

For mice immunization, we purified and separated VLPs through a 20% sucrose cushion. Since non-VLP viral protein cannot pass through the 20% sucrose cushion, we think that there is no contaminations of non-VLP viral protein.

Moreover, since this journal gives a relatively short revision period, further experiment is not possible. Please understand. 

Once again, I appreciate for your time and comments.

Reviewer 2 Report

The authors respond to my concerns. I do not have any further comments.

Author Response

I appreciate for your time and comments.

Reviewer 3 Report

The authors reviewed and documented the review’s suggestions. The manuscript improved significantly.   

Author Response

(The authors gave the same response as above.)

Round 3

Reviewer 1 Report

I agree it is a special moment, so I gave accept.